# FULLY DIFFERENTIABLE FULL-ATOM PROTEIN BACKBONE GENERATION

**Namrata Anand**
Bioengineering Department, Stanford
`namrataa@stanford.edu`

**Raphael Eguchi**
Biochemistry Department, Stanford
`reguchi@stanford.edu`

**Po-Ssu Huang**
Bioengineering Department, Stanford
`possu@stanford.edu`

## ABSTRACT

The fast generation and refinement of protein backbones would constitute a major advancement to current methodology for the design and development of *de novo* proteins. In this study, we train Generative Adversarial Networks (GANs) to generate fixed-length full-atom protein backbones, with the goal of sampling from the distribution of realistic 3-D backbone fragments. We represent protein structures by pairwise distances between all backbone atoms, and present a method for directly recovering and refining the corresponding backbone coordinates in a differentiable manner. We show that interpolations in the latent space of the generator correspond to smooth deformations of the output backbones, and that test set structures not seen by the generator during training exist in its image. Finally, we perform sequence design, relaxation, and *ab initio* folding of a subset of generated structures, and show that in some cases we can recover the generated folds after forward-folding. Together, these results suggest a mechanism for fast protein structure refinement and folding using external energy functions.

## 1   INTRODUCTION

Deep generative models, which harness the power of deep neural networks, have achieved remarkable results in realistic sample generation across many modalities including images (1; 2; 3; 4; 5), video (6; 7), audio (8), and symbolic expressions (9). These methods have been further applied to problems in biology and chemistry, such as the generation of small molecules (10) and more recently, protein backbones (11). The ability to easily sample from the distribution of viable proteins would be useful for the development of new therapeutics, where often the goal is to determine the structure of a putative ligand for a known target receptor or to realistically modify an existing protein. As a more general engineering problem, speeding up and improving the *de novo* protein design process would be extremely valuable in the modeling and development of new biosensors, enzymes, and therapeutics.

Recently, Generative Adversarial Networks (GANs) were trained to generate matrices ("maps") representing pairwise distances between alpha-carbons on fixed-length protein backbones (11). This representation of the backbone is rotationally and translationally invariant and captures long-range 3-D contacts; training on these maps allows for the stable generation of highly varied structures. However, a drawback to the approach in (11) is that the underlying 3-D coordinates of the backbone must be recovered and local errors in the backbone due to errors in the generated maps must be corrected. The reported method for coordinate recovery in (11) is not differentiable and requires iterative optimization.

In this study, we extend the methods in (11) by (i) generating *full-atom* pairwise distance matrices for fixed-length fragments and (ii) training deep neural networks to recover and refine the corresponding coordinates. Importantly, we show that for a subset of structures, we can design sequences onto the generated backbones, and recover their structures by forward-folding using the Rosetta macro-molecular design suite (12; 13). Our results suggest that a subset of the generated fragments can host folding sequences and thus are viable starting scaffolds for *de novo* protein design. Our goal is

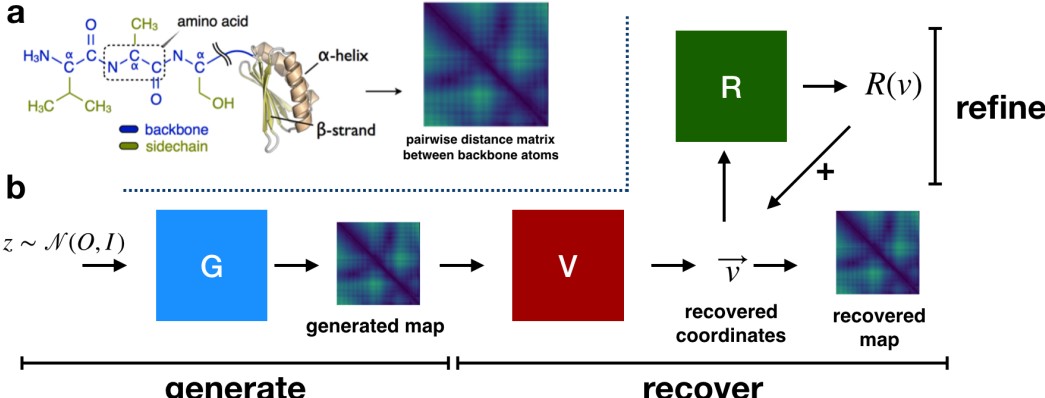

Figure 1: **a) Data representation**. We represent protein structures using pairwise distances in angstroms between all of the atoms on the protein backbone. Figure adapted from (11). **b) Pipeline**. The generator $G$ **generates** a pairwise distance matrix, for which the underlying coordinates are **recovered** by network $V$. Further coordinate **refinement** is done with additive updates by network $R$.

to eventually incorporate external heuristic energy functions into the learning algorithm, to further refine the generated backbones.

## 2 BACKGROUND

**Protein backbone structure**. Proteins are macromolecules comprised of sequences of amino acids, which are polymerized via a condensation reaction between the amine and carboxylic acid groups of each amino acid (Figure 1a). The protein "backbone" refers to the main chain of the protein, which consists of the alpha-carbon ($C_\alpha$) of the amino acid, as well as the other atoms constituting the peptide bond. The backbone has a repeating pattern of nitrogen, carbon, and oxygen atoms, and the torsion of the backbone follows a fixed distribution, with angles $\phi, \psi$, and $\omega$ defined as the angles around the $N - C_\alpha$, $C_\alpha - C$, and $C - N'$ bonds, respectively. Each amino acid is chiral, and as a result, the global structure of the protein assumes a particular "handedness" that is reflected in the torsion angle distribution. At the secondary structure level, this leads to phenomena such a the formation of right-handed helices with naturally occurring amino acids.

While protein side-chains, and hence the amino acid sequence, dictate the structure of a protein, generating sequences to fit an existing backbone (sequence design) has become common practice in recent years (14). The greatest challenge in protein design lies instead in the construction of new backbones for which sequences can be designed; that is, after sequence design, the predicted structure from sequence alone should correspond to the original backbone. The ability to reliably generate these "foldable" structures is what inspires our undertaking.

In this study, we demonstrate that deep generative models can be used to approximate the distribution of real protein backbones, and delineate a process by which such models can be used for *de novo* design of proteins.

**Related Work.** Current state-of-the-art methods for backbone generation rely on either sampling or assembling native backbone fragments (15; 16). This study directly extends the method presented in (11), which baselines alpha-carbon pairwise distance matrix generation against other existing statistical methods for protein structure generation (17; 18). We improve upon the full-atom GAN baseline presented in (11) and present a fast, fully differentiable method for recovering the underlying coordinates corresponding to a generated matrix.

By "fully differentiable," we refer to the fact that errors with respect to the 3-D backbone coordinates generated by our pipeline can backpropagate to the generator network. There are clearly other methods for protein backbone generation which are by construction differentiable– for example, generating backbone torsion angles or directly generating coordinates. However, in the former case, we found that generated torsion angles often lacked reasonable 3-D structure, with structures being highly distended or "tangled" in a way that violates realistic peptide geometries. In the latter case, models suffered from mode collapse, even in the case when the discriminator module acts on pairwise distance matrices, possibly due to the coordinate space representation not being invariant to rotations and translations.

## 3 EXPERIMENTAL SETUP

An overview of our method is shown in Figure 1b. Our experimental pipeline consists of three phases: we **generate** pairwise distances corresponding to a 64-residue full-atom protein backbone, **recover** the corresponding 3-D coordinates, and finally, **refine** the recovered coordinates to reduce local errors in the backbone. Here we train each module separately, but these can be trained or fine-tuned end-to-end.

### 3.1 GENERATE

We first train a Generative Adversarial Network (GAN) (1) to **generate** pairwise distance matrices for full-atom 64-residue fragments ($256 \times 256$ inputs). The GAN is comprised of two deep neural networks – a generator and a discriminator. The generator network $G$ attempts to produce realistic data, while the discriminator network $D$ attempts to distinguish between real samples $\mathbf{x}$ and the fake samples generated by $G$. Their individual objectives are

$$\max_D \; \mathbb{E}_{\mathbf{x} \sim p_{\text{data}}(\mathbf{x})}[\log D(\mathbf{x})] + \mathbb{E}_{\mathbf{z} \sim p_{\mathbf{z}}(\mathbf{z})}[\log(1 - (D(G(\mathbf{z}))))]$$

$$\max_G \; \mathbb{E}_{\mathbf{z} \sim p_{\mathbf{z}}(\mathbf{z})}[\log(D(G(z)))]$$

(1)

where $\mathbf{z} \sim \mathcal{N}(0, I)$ and the range of the discriminator $D$ is [0,1].

As in (11), we clamp the generated maps above zero and symmetrize them. We use a DCGAN architecture (4) with convolution transpose operations for upsampling by the generator. We use a noise vector size of 1024 units and train our models using the Adam optimizer (19) ($\beta_1 = 0.5, \beta_2 = 0.999$) with generator and discriminator base learning rates set to $7.5 \times 10^{-5}$ and $10^{-5}$, respectively. We train with a batch size of 8 and reduce the learning rates by a factor of 10 every 25K iterations.

### 3.2 RECOVER

We then **recover** coordinates corresponding to generated maps by (i) pre-training a coordinate recovery network on real maps and (ii) fine-tuning the network on generated maps.

**Pre-training on real data**. We found that a deep convolutional neural network could be trained to recover protein backbone coordinates from pairwise distance matrices via an autoencoder loss. This is somewhat surprising, since for any input matrix, there are an infinite number of correct coordinates corresponding corresponding to arbitrary rotations and translations. However, the trained network seems to have underlying "logic" for coordinate placement; for example, the recovered coordinates tend to be centered about the origin. We first pre-train a network $V$ on the task of recovering coordinates $\vec{v} = V(\hat{m})$ from real input distance maps $\hat{m}$. We train with a scaled autoencoder loss $L_{AE}$.

$$L_{AE}(\hat{m}) = \frac{(map(V(\hat{m})) - \hat{m})^2}{\hat{m}^2}$$

(2)

where the *map* operation converts generated coordinates $\vec{v} = V(\hat{m})$ via construction of the Gram matrix.

This objective is not sufficient for learning to recover backbone coordinates. Since optimal coordinate recovery from pairwise distances is only identifiable up to reflection of the coordinates, we need

additional constraints in order to recover the correct handedness of the protein backbone. We therefore add a rigid body alignment loss $L_{RB}$

$$L_{RB}(\hat{m}) = \lambda \, \|R \, V(\hat{m}) + t - \hat{m}\|_2 \tag{3}$$

where rotation matrix $R$ and translation vector $t$ are selected to minimize the least squares loss. These can be found in one step via singular value decomposition (20). In practice, we set $\lambda = 1e - 2$.

**Fine-tuning on generated maps**. We then fine-tune the pre-trained coordinate recovery network on generated maps, while continuing to train on real data. For generated maps, there are no ground truth coordinates to align to (no equivalent $L_{RB}$), and simultaneously training on real data is not sufficient to ensure the correct handedness of recovered coordinates. Therefore, we introduce the following two discriminator networks.

- **Local block discrimination**. The first discriminator network $D_B$ operates on strided on-diagonal blocks of the input map, discriminating between real and generated data. This network corrects the torsion of the recovered backbone locally, but cannot ensure the chirality of the backbone is correct. We select a block size of $16 \times 16$ with a stride of 4.
- **Local fragment discrimination**. The second network $D_F$ discriminates between real and fake excised fragments from recovered coordinates. These fragments are canonicalized in the following manner: each fragment's center of mass is set to the origin, and the fragment is aligned along the positive $z$-axis from N- to C- terminus. This alignment is done by finding the first principal component for the fragment coordinates, and aligning that vector to the $z$-axis; the sign of the vector is checked to ensure that the fragment lies in the direction of increasing $z$ from N- to C- terminus. This network ensures that the local chirality of recovered coordinates is correct. We train $D_F$ with a fragment length of 16 atoms, with a stride of 4.

While fine-tuning the coordinate recovery network $V$ on generated maps, the network learns to recover coordinates that match the input pairwise distances but also that fool these two discriminators, as in typical GAN training. The final training loss objectives for $V$ on real and generated maps are

$$\begin{aligned} L_{\text{real}}(\hat{m}) &= L_{AE}(\hat{m}) + L_{RB}(\hat{m}) \\ L_{\text{fake}}(m) &= L_{AE}(m) + \gamma \log[1 - D_F(\textit{frag}(V(m)))] + \gamma \log[1 - D_B(\textit{block}(V(m)))] \end{aligned} \tag{4}$$

where *block* and *frag* corresponds to the local block extraction and fragment canonicalization procedures described above. $V$, $D_F$, and $D_B$ are deep convolutional networks, and we set $\gamma = 0.04$ in practice. We pre-train $V$ with fixed learning rate $5 \times 10^{-4}$ and fine-tune $V$, $D_F$, and $D_B$ with fixed learning rate $10^{-4}$, all with batch size 8.

### 3.3   REFINE

After training the coordinate recovery module, we found that recovered structures often had local backbone errors which could be improved. We have two candidate **refinement** processes to correct these errors.

**Refinement network.** Given fixed input coordinates $\vec{v}$, we train a refinement network $R$ to learn an additive shift $R(v)$. We unroll this operation for $t$ steps, with the recurrence $\vec{v}_t = \vec{v}_{t-1} + R(\vec{v}_{t-1})$. As with the coordinate recovery module, we train a local block discriminator $D_B$ and a fragment discriminator $D_F$, and we update the network $R$ with an autoencoder loss and discriminator fooling loss at the last time step:

$$L_t(\vec{v}) = \|\vec{v}_t - \vec{v}_0\|_2 + \gamma \log[1 - D_F(\textit{frag}(\vec{v}_t))] + \gamma \log[1 - D_B(\textit{map}(\vec{v}_t))] \tag{5}$$

where $V$ and $G$ are kept fixed, and $D_B$ and $D_F$ are trained in the same way as previously described. We train $R$ with fixed learning rate $10^{-4}$ and unroll the recurrence for 5 time steps during training and testing.

**Meta-learning.** While our trained coordinate recovery network $V$ is not able to generalize to arbitrary generated maps without local backbone errors, the network itself is a good "local minimum" in that it

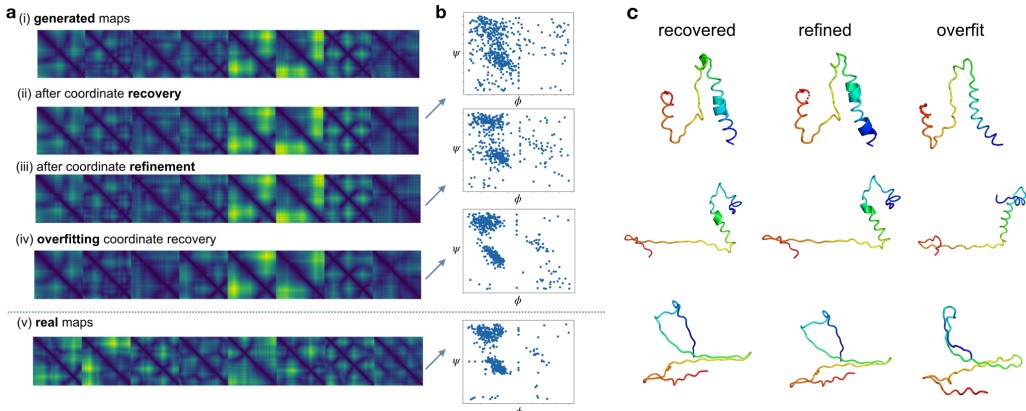

Figure 2: **a) Qualitative features of generated maps are preserved after structure recovery**. We render (i) random generated pairwise distance matrices ("maps"), as well as maps after (ii) coordinate recovery, (iii) coordinate refinement, and (iv) overfitting, along with (v) real maps. **b)** $\phi, \psi$ **torsion distribution post- recovery and refinement**. Torsion distribution after coordinate recovery indicates that recovered structures have the correct handedness. Coordinate refinement improves the torsion distribution after coordinate recovery. (Distribution of $\omega$ angles omitted; recovery matches true distribution well) **c) Recovered and refined fragments**. Random generated fragments after coordinate recovery (left). Refinement either by learned refinement module (middle) or via overfitting coordinate recovery (right)

can quickly learn to recover coordinates well for a single or small batch of examples. We overfit in this way to new generated maps using the pretrained coordinate recovery network $V$ as well as the pretrained discriminators $D_B$ and $D_F$ and find that we can recover coordinates which adhere well to the true torsion distribution for these specific examples. When overfitting, we set the learning rate of $V$ to $10^{-3}$ and overfit for 3K iterations.

## 4 EXPERIMENTS

**Dataset**. Our training data consists of protein crystal structures from the Protein Data Bank (21). Our dataset is identical to that described in (11), except rather than use $\alpha$-carbon pairwise distance matrices for fragments, we use full-atom pairwise distances. Our final dataset is made up of 800K fragments taken from the 'A' chain of structures in the PDB. We omit structures with backbone atomic bonds which do not adhere to within 1 Å of average idealized bond lengths (22).

### 4.1 FULL-ATOM BACKBONE GENERATION

In Figure 2a, we show random full-atom maps generated by our trained GAN. These full-atom maps closely resemble real maps, and coordinate recovery and refinement do not significantly alter the qualitative features of the generated maps. Importantly, the torsion distribution after coordinate recovery is realistic and indicates that the recovered structures have the right handedness (Figure 2b). The learned refinement module and the meta-learning/overfitting procedure both improve the torsion distribution of the generated structures, without introducing large deviations from the original generated distance constraints. We also render random generated 64-residue protein fragments, after coordinate recovery and refinement (Figure 2c). Although the overfitting procedure produces the most realistic backbones in terms of local backbone structure, in practice, we would prefer to be able to generate structures with a single forward pass rather than requiring additional optimization for each structure.

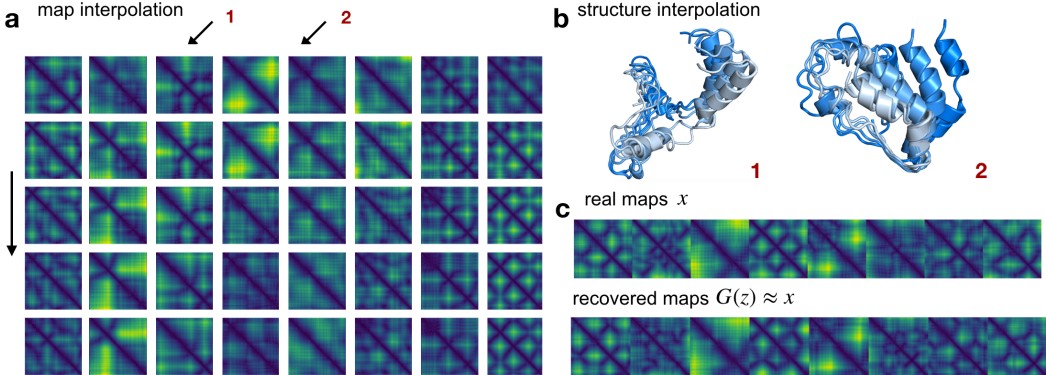

Figure 3: **a) Smooth map interpolation**. Linear interpolation between sampled latent vectors corresponds to smooth interpolation of output pairwise distance matrices (top to bottom). **b) Smooth structure interpolation**. Recovered and refined structures corresponding to maps in (a) (arrows). Linear interpolation of $z$ corresponds to smooth deformations of structure (light to dark). **c) Latent space recovery of test set structures**. For given test set maps $x$, we optimize $z$ to find $G(z) \approx x$.

## 4.2 Examining the capacity of the generative pipeline

Ultimately, we would like to be able to utilize heuristic energy functions to guide learning for our generative pipeline. To assess our current pipeline's capacity for fast protein structure refinement and folding using external energy functions, we ask the following questions:

- Do small steps in the latent space of the generator correspond to realistic deformations of the protein backbone?
- Are unseen real structures expressible by the generator?
- Can we design sequences onto some of the generated backbone fragments and subsequently forward fold the sequences into the same structure?

Together, adhering to these criteria ensures that coordinates can be updated smoothly with respect to error signals from heuristic energy functions, that the generator is suitably expressive enough in that it contain native folds in its image, and finally, that the generated structures are actually viable scaffolds for *de novo* protein design.

**Structure interpolation.** We first asked whether linear interpolations in the latent space of the generator correspond to smooth structural deformations of the peptide backbone after coordinate recovery $(V)$ *and* refinement $(R)$. In Figure 3a, we render the output of the generator $G(z)$ while linearly interpolating in $z$ (top to bottom); we see that map interpolations are smooth. In addition, the corresponding structures after coordinate recovery $V$ and 5 steps of refinement $R$ deform smoothly (Figure 3b), and intermediate interpolated structures retain connectivity of the peptide backbone as well as realistic secondary structure.

**Structure finding.** We then asked whether for a given test set pairwise distance matrix $x$, we can find a corresponding latent vector $z$ such that $G(z) \approx x$, where $G$ is the pre-trained generator network. We optimized $z$ using the same method presented in (11), where we minimize the L2 distance between the generated map $G(z)$ and the real example $x$, along with a K-L divergence regularizer term on $z$. We optimized $z$ for 5K iterations using the Adam optimizer with learning rate $5 \times 10^{-2}$ and batch size 8. In Figure 3c, we render an example of structure finding for a few input examples; we find that the generator is expressive enough such that real unseen data examples exist in the image of the generator.

**Sequence design and *ab initio* folding of generated structures.** While there are infinitely many coordinate sets that satisfy the properties of peptide backbones, only a small fraction of these are able to host an amino acid sequence that folds into the specified structure. Creating viable backbones

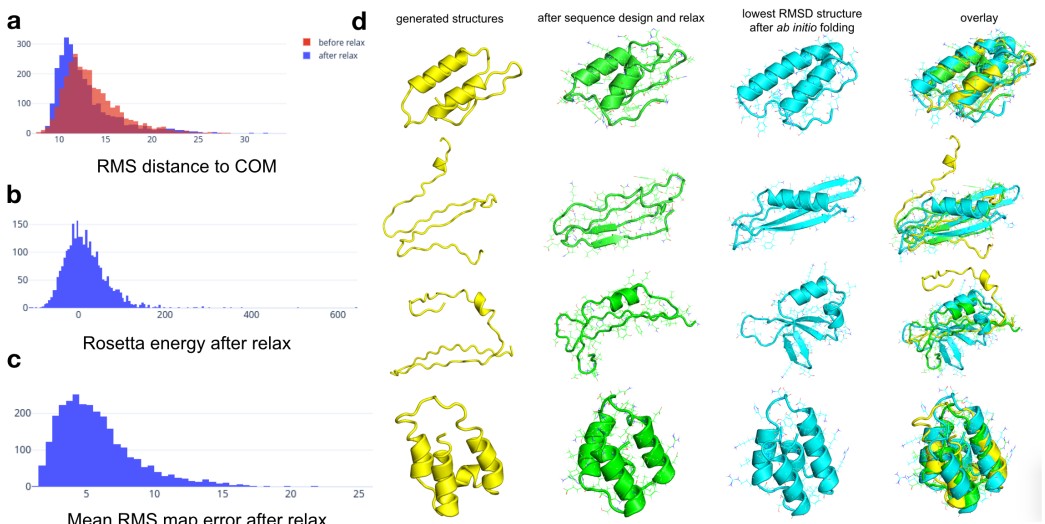

Figure 4: **Sequence design and *ab initio* folding of generated structures** 2800 structures are sampled from the generator. **a) Radius of gyration (RoG) –** Mean RMS distance to center-of-mass across generated structures before (red) and after (blue) relax. Only few of the structures are distended (long tail of the distribution). **b) Rosetta energy** of structures after sequence design and relax. **c) Mean RMS map error after relax**. Note that after relax, we expect deviation from the original distance constraints. **d) *ab initio* folding** of candidate low RoG generated and refined structures with low Rosetta energy after sequence design. Lowest RMSD structure to input relaxed structure rendered. In many cases the designed sequence refolds to a similar global structure, while preserving the input secondary structure.

remains a challenge in the protein design field, and state-of-the-art protocols (12; 23) still rely on partial structures harvested from the Protein Data Bank to construct designable backbones.

To test whether generated backbones can host foldable sequences, we randomly sampled 2800 structures and performed 3 rounds of iterative sequence design and structural refinement using Rosetta (24; 25). Because the generative model is trained on fragments, we do not expect in general that generated structures should adopt globular shapes. In this application, the generator thus provides orderings of secondary structure elements encoded in local pairwise distance constraints – a non-trivial task in backbone design (26; 27) – while Rosetta is used to improve the rough global positioning of each secondary structure element. Movements of secondary structure elements during the relax step are therefore both anticipated and desired.

Following the relax step, roughly 40% of structures achieved negative Rosetta energy scores (Figure 4b), with movements in the backbone (Figure 4c) allowing for the formation of internal contacts that result in lower energies (Figure 4d). Designs with low radii of gyration (Figure 4a) and favorable (*viz.* negative) Rosetta energies were selected for further testing.

To assess the viability of these models, we tested whether select low-energy, designed structures could be recovered by forward-folding using *Rosetta AbInitio*, which performs blind structure prediction from the designed sequence alone. The results are shown in Figure 4. While the sequence design process resulted in some deviations from the generated models, torsional and secondary structure patterns were noticeably conserved, and the designed models were closely recovered in forward-folding (Figure 4d). These results suggest that our generator is capable of providing designable peptide backbones, the viability of which is supported both by favorable Rosetta energy values for designed structures, as well as by the ability to perform blind structure recovery from sequence.

## 5 CONCLUSION

In this paper, we propose a pipeline for generating fixed-length full-atom protein backbones in a fully differentiable manner. We train a model to generate pairwise distance matrices between atoms on the backbone, which eliminates the need to explicitly encode structure invariances to arbitrary rotations and translations, while also modeling long-range contacts. We show that we can then train networks to learn to recover and refine the underlying coordinates.

Finally, we take steps to show the capacity of our pipeline for *de novo* protein design. Specifically we show that interpolations in the latent space of the generator correspond to smooth deformations of the recovered peptide backbone, that native unseen structures exist in the image of the generator, and that a subset of generated backbones can host foldable sequences. Together, these results suggest a mechanism for fast protein structure refinement and folding using external energy functions.

We plan next to learn to generate relaxed, low-energy structures by directly optimizing our generative pipeline using the Rosetta energy function (23), differentiating through the coordinate recovery and refinement modules. We plan to further extend our work by generating longer or arbitrary length backbones, as well as by conditioning our generative model on secondary structure, so that we can specify backbone topologies for design.

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
