# OpenReview forum: "Fully differentiable full-atom protein backbone generation"
_ICLR.cc/2019/Workshop/DeepGenStruct — DeepGenStruct 2019_

### Official Review · AnonReviewer2 · 2019-04-04
**Interesting Graph Generation Paper**

**Rating:** 4
**Confidence:** 2

**Review:**

This work presents a generative model for protein backbones (graphs). A GAN is used to generate a map of all pairwise distances among nodes. Then, an autoencoder-like network tries to place the nodes in the 3D space. Finally a refinement process is used to improve the output. A set of qualitative evaluations suggest positive results.

I believe that this paper is interesting to be accepted in the workshop.

Overall, many aspects of the paper are related to the application of protein folding that I feel unqualified to judge. I would have found useful a better introduction to the problems in the area to understand the application.

* The evaluation seems interesting but I would have hoped for a more quantitative evaluation. Although I do not know of the domain-specific metrics, a common theme in such papers is to compare the probability distributions of some extrinistic characteristics of molecules/proteins between sample in the dataset and generated samples. The current qualitative results are nice and reasonable for a workshop paper, but this leaves no way for future work to compare to this one.

* Currently, the generation of the distance maps, the recovery network and the refine modules are trained separately. It would be nice if the authors could explain why they avoided an end-to-end training procedure. Is there an optimization issue? Is this somehow related to the application?

* The generated proteins are of fixed length (64). I assume that this isn't true of all proteins. (a) How does this method scale with the size of these proteins? (b) Does the current framework allow for variable-sized graphs?

---

### Official Review · AnonReviewer1 · 2019-04-14
**Nice application of GANs to protein generation.**

**Rating:** 4
**Confidence:** 2

**Review:**

This paper presents an end-to-end approach for generating protein backbones using generative adversarial networks, an important direction in applying deep generative models for generating complex, highly-structured biological data (proteins).  First, a generative model (DCGAN) is used to generate pairwise distance matrices between atoms on the backbone. Second, a recover network transforms the distance matrix into the underlying 3D coordinates of the backbone. Finally, the recovered coordinates are post-edited by a recurrent error correction model.

In the experimental section, the authors present a rich set of qualitative evaluations, with examples and case studies demonstrating that the underlying latent space is smooth, and small steps in the latent space of the generator correspond to realistic deformations of the protein backbone. Unseen test examples are also encoded in the latent space. The authors conclude the paper by outlining an approach to iteratively refined generative backbones to host foldable sequences.

The paper is clearly written and the method is novel. My only comment is that it seems the submission lacks quantitive analysis, with most conclusions, were drawn from illustrative case studies. Additionally, the authors could also try using the Rosetta energy function as part of the discriminator, directly optimizing the generated structures to have low energy.

---

### Decision · Program_Chairs · 2019-04-19
**Acceptance Decision**

Accept